# CARVE: Dissecting Core Components for Accurate and Resolution-Enhanced Visual Geometry Estimation

## Abstract

Despite significant advances in feed-forward visual geometry estimation, several core components that underpin model performance remain underexplored. In this work, we systematically dissect the core components and enhance the performance of VGGT Wang et al. (2025a). Our ablation experiments reveal that: 1) Data diversity remains a more impactful factor for accuracy than data quality; 2) The widely adopted confidence-aware loss and spatial gradient loss can unexpectedly degrade performance. We further evaluate the effectiveness of several existing techniques, demonstrating that sequence-level and frame-level alignment improve overall performance, while local region alignment unexpectedly brings a performance drop. In addition, we propose two enhancements: a consistency loss that enforces coherence among depth maps, camera parameters, and point maps; and an efficient architectural adaptation that enables high-resolution visual geometry estimation. These insights and improvements are integrated into **CARVE**, a model that jointly predicts accurate and consistent geometry from arbitrary input views with high resolutions. Extensive experiments for point cloud estimation, video depth estimation, and camera pose and intrinsic estimation across diverse benchmarks demonstrate that CARVE achieves state-of-the-art performance in visual geometry estimation.

## 1 Introduction

Recovering accurate and consistent 3D attributes from monocular video, including 3D point clouds, camera poses, intrinsic parameters, and depth maps, remains a long-standing challenge with broad applications, such as autonomous driving Chen et al. (2020); Zhu et al. (2012), virtual and augmented reality Mahmood et al. (2020); Placitelli & Gallo (2011), robotic navigation Wang et al. (2021); Liu (2015), and medical imaging Starly et al. (2005); Cheng et al. (2021).

Existing approaches can be roughly categorized into two groups: optimization-based approaches and learning-based approaches. Optimization-based techniques Schonberger & Frahm (2016); Cui et al. (2017); Pan et al. (2024); Moulon et al. (2013) generally rely on robust feature extraction and matching to estimate 3D attributes by minimizing reprojection errors. These methods often produce sparse or semi-dense reconstructions due to the heavy dependence on reliable correspondences.

In contrast, learning-based approaches directly regress 3D attributes through end-to-end neural networks trained on large-scale labeled datasets, which can be categorized into per-frame approaches Yang et al. (2024b); Hu et al. (2024); Bochkovskii et al. (2024); Wang et al. (2025c) and multi-frame methods Wang et al. (2025a; 2024); Leroy et al. (2024); Wang & Agapito (2025); Yang et al. (2025); Wang et al. (2025b). Interestingly, per-frame methods often exhibit weaker multi-frame consistency but superior accuracy on individual frames. This performance gap is counterintuitive because multi-frame models can theoretically leverage additional multi-frame matching information.

Recently, VGGT Wang et al. (2025a), a representative multi-view transformer architecture, has attempted to bridge this gap by introducing alternating-attention mechanisms. This architecture integrates inter-frame and global self-attention modules that can enhance both per-frame accuracy and multi-frame consistency theoretically. However, the aforementioned performance gaps remain non-negligible. On the other hand, per-frame methods such as MoGe Wang et al. (2025c) and Depth

Pro Bochkovskii et al. (2024) attribute their strong single-frame performance to carefully designed training objectives, high-resolution inputs, and training curriculum. Motivated by these observations, we systematically investigate these key factors on VGGT through extensive ablation studies, resulting in four critical insights: 1) Despite extensive pre-training of VGGT on large datasets, training data diversity plays a more vital role for accuracy than data quality; 2) The commonly adopted spatial gradient loss and confidence-aware weighting strategy can unintentionally degrade the model performance. Conversely, employing fixed weighting inversely proportional to ground-truth depth consistently improves performance; 3) The sequence-level and frame-level alignment strategy of training objectives improves overall performance, whereas the local region alignment unexpectedly brings a performance reduction.

In addition, we further introduce two novel enhancements to integrate the advantage of optimization-based methods and leverage high-resolution inputs. First, motivated by the intrinsic geometric constraints of optimization-based techniques, we introduce a consistency loss enforcing strict coherence among estimated camera parameters, depth maps, and 3D point clouds. Second, rather than directly feeding high-resolution inputs into VGGT, we propose to extract separate high-resolution and low-resolution ViT features, and fuse them via cross-attention modules equipped with zero-initialized gating parameters, which can preserve the VGGT pre-training knowledge.

Combining these insights and methodological improvements, we scale up the training and propose **CARVE**, an accurate and resolution-enhanced visual geometry estimation model. The rigorous ablation study paves the way for state-of-the-art performance of video depth estimation, camera pose and intrinsic estimation, and 3D point cloud estimation on evaluation datasets including KITTI Geiger et al. (2013), 7-Scenes Shotton et al. (2013), TUM Sturm et al. (2012), HO3D Hampali et al. (2020), ETH3D Schops et al. (2017), HAMMER Jung et al. (2023), and Bonn Palazzolo et al. (2019). Our main contributions are summarized as follows:

- Rigorous ablation study experiments are conducted on VGGT to explore how training objectives and data influence the visual geometry performance.
- We propose a novel consistency loss to enforce geometric coherence among predicted camera parameters, depth maps, and 3D point clouds, thereby integrating intrinsic perspective projection constraints during training.
- We develop an efficient and effective feature fusion mechanism that integrates high-resolution features into low-resolution features via cross-attention, enabling accurate estimation from high-resolution inputs with less computational burden.
- Integrating these insights and improvements, we propose the **CARVE** framework, achieving new state-of-the-art results in video depth estimation, camera pose and intrinsic parameter estimation, and point cloud estimation across diverse benchmark datasets.

## 2 RELATED WORK

### 2.1 OPTIMIZATION-BASED RECONSTRUCTION

Traditional Structure-from-Motion (SfM) Moulon et al. (2013); Schonberger & Frahm (2016); Cui et al. (2017); Pan et al. (2024), Multi-View Stereo (MVS) Schönberger et al. (2016); Goesele et al. (2007); Furukawa & Ponce (2009), and visual SLAM methods Mur-Artal et al. (2015); Engel et al. (2014; 2017) extract artificial features, match them between frames, and optimize to minimize the reprojection error to estimate camera poses and reconstruct 3D geometry. In order to enhance robustness, several methods Teed & Deng (2021); Li et al. (2025); Yao et al. (2018); Huang et al. (2018a); Ding et al. (2022) incorporate learning-based features or correspondence information to assist in the optimization process.

### 2.2 PER-FRAME RECONSTRUCTION

Numerous studies have investigated monocular depth estimation, demonstrating notable progress and improved accuracy Yang et al. (2024a;b); Ke et al. (2024); Yin et al. (2023); Hu et al. (2024); Bochkovskii et al. (2024); Yin et al. (2021). Several of them Yin et al. (2023); Bochkovskii et al. (2024) can produce 3D reconstruction from a single image. Moge series Wang et al. (2025c;d) propose to directly estimate dense affine-invariant point clouds. To ensure consistency between

frames, they still rely on alignment with matching information Sun et al. (2021); Sarlin et al. (2020) or consistency optimization Xu et al. (2023).

### 2.3 MULTI-FRAME RECONSTRUCTION

Recently, based on previous matching-based methods Chang & Chen (2018); Liu et al. (2022); Cheng et al. (2020), DUSt3R Wang et al. (2024) and its subsequent work Leroy et al. (2024); Jang et al. (2025); Zhang et al. (2025) propose to directly learn point clouds from the input images. Furthermore, Spann3R and CUT3R Wang & Agapito (2025); Wang et al. (2025b) extends the input to longer temporal sequences employing implicit scene representations. Recently, Fast3R Yang et al. (2025) and VGGT Wang et al. (2025a) utilize a multi-view transformer to process features from all images in a single forward pass to estimate long-sequence geometric structures directly.

## 3 METHOD

In this section, we revisit the visual geometry estimation framework VGGT Wang et al. (2025a) and systematically investigate the effects of training data composition, training objectives, and input resolution. Considering these key aspects, we propose a consistency loss and an efficient high-resolution adaptation strategy. Building on these advances, we scale up training and introduce CARVE, a robust model that achieves state-of-the-art performance in visual geometry estimation.

### 3.1 PRELIMINARIES: VGGT

**Problem Definition.** VGGT Wang et al. (2025a) takes a set of images $\mathbf{I} \in \mathbb{R}^{T \times H \times W \times 3}$ as input, and produces depth maps $\hat{\mathbf{D}} \in \mathbb{R}^{T \times H \times W}$, world coordinates point maps $\hat{\mathbf{P}} \in \mathbb{R}^{T \times H \times W \times 3}$, and camera parameters $\hat{\mathbf{g}} \in \mathbb{R}^{T \times 9}$, which are composed of quaternion $\hat{\mathbf{r}} \in \mathbb{R}^{T \times 4}$, translation vectors $\hat{\mathbf{t}} \in \mathbb{R}^{T \times 3}$, and field of view angels $\hat{\boldsymbol{\theta}} \in \mathbb{R}^{T \times 2}$. The point tracking task is not discussed here.

**Network Architecture.** VGGT patchifies the input images $\mathbf{I}$ into tokens $\hat{\mathbf{f}}_{\text{img}} \in \mathbb{R}^{T \times P \times C}$ with DINOv2 Oquab et al. (2023) encoder and then passes them along with learnable camera tokens $\mathbf{f}_{\text{cam\_init}}$ into transformer blocks and decoders.

$$\hat{\mathbf{f}}_{\text{img}} = \text{Encoder}(\mathbf{I}), \; (\hat{\mathbf{f}}_{\text{geo}}, \hat{\mathbf{f}}_{\text{cam}}) = \text{Transformer}(\hat{\mathbf{f}}_{\text{img}}, \mathbf{f}_{\text{cam\_init}}),$$
$$\hat{\mathbf{D}} = \text{Head}_{\text{depth}}(\hat{\mathbf{f}}_{\text{geo}}), \;\; \hat{\mathbf{P}} = \text{Head}_{\text{point}}(\hat{\mathbf{f}}_{\text{geo}}) \tag{1}$$
$$\hat{\mathbf{g}} = [\hat{\mathbf{t}}, \hat{\mathbf{r}}, \hat{\boldsymbol{\theta}}] = \text{Head}_{\text{cam}}(\mathbf{f}_{\text{cam}}).$$

**Training Objectives.** The training losses contain three types of functions. For regression loss, they filter out invalid regions and simply supervise the valid regions:

$$\mathcal{L}_{\text{reg}}(\hat{\boldsymbol{\xi}}, \boldsymbol{\xi}, \mathbf{W}) = \mathbb{E}_{p \in \mathcal{M}} \left\| \mathbf{W}_p \cdot (\hat{\boldsymbol{\xi}}_p - \boldsymbol{\xi}_p) \right\|, \tag{2}$$

where $\hat{\boldsymbol{\xi}}$ and $\boldsymbol{\xi}$ are the prediction and the ground truth of either depth maps or point maps. $\mathcal{M}$ represents the valid region, and $\mathbf{W}$ means the weight map. For spatial gradient loss, it supervises the difference between nearby pixels:

$$\mathcal{L}_{\text{sg}}(\hat{\boldsymbol{\xi}}, \boldsymbol{\xi}, \mathbf{W}) = \mathbb{E}_{p \in \mathcal{M}} \left\| \mathbf{W}_p \cdot (\nabla_p \hat{\boldsymbol{\xi}}_p - \nabla_p \boldsymbol{\xi}_p) \right\|, \tag{3}$$

where $\nabla_p$ represents the difference between nearby pixels of the spatial x and y axes. Another confidence loss is adopted to supervise the learnable confidence map:

$$\mathcal{L}_{\text{conf}}(\mathbf{W}) = \mathbb{E}_{p \in \mathcal{M}} \left| -\alpha \log \mathbf{W}_p \right|, \tag{4}$$

The overall training losses consist of three components:

$$\mathcal{L} = \mathcal{L}_{\text{cam}} + \mathcal{L}_{\text{depth}} + \mathcal{L}_{\text{point}}, \;\; \mathcal{L}_{\text{cam}} = \mathcal{L}_{\text{reg}}(\hat{\mathbf{g}}, \mathbf{g}),$$
$$\mathcal{L}_{\text{depth}} = \mathcal{L}_{\text{reg}}(\hat{\mathbf{D}}, \mathbf{D}, \boldsymbol{\Sigma}^{\text{depth}}) + \mathcal{L}_{\text{sg}}(\hat{\mathbf{D}}, \mathbf{D}, \boldsymbol{\Sigma}^{\text{depth}}) + \mathcal{L}_{\text{conf}}(\boldsymbol{\Sigma}^{\text{depth}}) \tag{5}$$
$$\mathcal{L}_{\text{point}} = \mathcal{L}_{\text{reg}}(\hat{\mathbf{P}}, \mathbf{P}, \boldsymbol{\Sigma}^{\text{point}}) + \mathcal{L}_{\text{sg}}(\hat{\mathbf{P}}, \mathbf{P}, \boldsymbol{\Sigma}^{\text{point}}) + \mathcal{L}_{\text{conf}}(\boldsymbol{\Sigma}^{\text{point}})$$

where the confidence maps $\boldsymbol{\Sigma}^{\text{depth}}$ and $\boldsymbol{\Sigma}^{\text{point}}$ are learned automatically, serving as adaptive weights for the loss functions.

## 3.2 EFFECTIVENESS OF TRAINING COMPONENTS

By default, we initialize the model with the VGGT Wang et al. (2025a) pretrained weights, freeze the ViT feature extractor, and train the remaining components. The predicted point cloud, depth map, and camera translation are aligned with the ground-truth values via a scale factor for each sequence before computing the loss. Training is conducted with a dynamic batch size with up to 24 frames, and evaluation is performed on uniformly sampled keyframes with up to 200 frames per video. More details are provided in the appendix. Results are reported in Table 1.

**Training Data.** We progressively expand the training data from "Data1" to "Data3", with the composition summarized in Table 2. The corresponding results in Table 1 demonstrate that the overall performance consistently improves as the training data grows. In particular, while "Data2" consists solely of high-quality datasets, "Data3" further incorporates noisy datasets but enhances diversity, and it yields superior overall performance. This finding suggests that relying exclusively on high-quality data is less effective than training on a more diverse dataset.

> **Finding 1.** 1) Expanding data diversity consistently leads to overall performance gains; 2) Compared with data quality, data diversity exerts a greater influence on model accuracy.

Table 1: Ablation study for training data, training loss, and high-resolution input. For training loss, we explore the effectiveness of the original VGGT losses ($\mathcal{L}_{\text{sg}}$, $\mathcal{L}_{\text{conf}}$), several losses adopted in state-of-the-art methods ($\mathcal{L}_{\text{tg}}$, $\mathcal{L}_{\text{F}}$, $\mathcal{L}_{\text{S}}$), and our proposed consistency loss ($\mathcal{L}_{\text{consis}}$). "Rank" represents the average rank value across all metrics. Rows in gray denote training with up to 12 frames (vs. the usual 24) and evaluation with up to 100 frames (vs. the usual 200) due to GPU memory constraint.

| Method | 7-Scenes | | | Bonn | | | KITTI | | | TUM | | | Rank↓ |
| --- | --- | --- | --- | --- | --- | --- | --- | --- | --- | --- | --- | --- | --- |
| | Recon C-L1↓ | Pose ATE↓ | Depth Rel↓ | Recon C-L1↓ | Pose ATE↓ | Depth Rel↓ | Recon C-L1↓ | Pose ATE↓ | Depth Rel↓ | Recon C-L1↓ | Pose ATE↓ | Depth Rel↓ | |
| Data1 | 0.056 | 0.079 | 0.070 | 0.051 | 0.064 | 0.049 | 0.281 | 1.411 | 0.085 | 0.040 | 0.090 | 0.049 | 2.600 |
| Data2 | 0.052 | 0.078 | 0.069 | 0.051 | 0.071 | 0.052 | 0.277 | 1.267 | 0.083 | 0.040 | 0.090 | 0.052 | 2.067 |
| (Our data) Data3 | 0.049 | 0.065 | 0.065 | 0.048 | 0.055 | 0.046 | 0.263 | 0.937 | 0.082 | 0.038 | 0.050 | 0.042 | **1.000** |
| (VGGT loss) $\mathcal{L}_{\text{reg}} + \mathcal{L}_{\text{conf}} + \mathcal{L}_{\text{sg}}$ | 0.049 | 0.065 | 0.065 | 0.048 | 0.055 | 0.046 | 0.263 | 0.937 | 0.082 | 0.038 | 0.050 | 0.042 | 2.000 |
| $\mathcal{L}_{\text{reg}} + \mathcal{L}_{\text{conf}}$ | 0.050 | 0.064 | 0.065 | 0.043 | 0.046 | 0.046 | 0.270 | 1.059 | 0.082 | 0.038 | 0.050 | 0.043 | 1.833 |
| $\mathcal{L}_{\text{reg}}(\mathbf{W}_{\text{inv}})$ | 0.043 | 0.066 | 0.062 | 0.044 | 0.050 | 0.046 | 0.254 | 0.866 | 0.079 | 0.036 | 0.039 | 0.039 | **1.333** |
| $\mathcal{L}_{\text{reg}}(\mathbf{W}_{\text{inv}})$ | 0.043 | 0.066 | 0.062 | 0.044 | 0.050 | 0.046 | 0.254 | 0.866 | 0.079 | 0.036 | 0.039 | 0.039 | 2.417 |
| $\mathcal{L}_{\text{reg}}(\mathbf{W}_{\text{inv}}) + \mathcal{L}_{\text{sg}}$ | 0.045 | 0.066 | 0.063 | 0.045 | 0.048 | 0.048 | 0.270 | 0.949 | 0.082 | 0.038 | 0.039 | 0.041 | 4.167 |
| $\mathcal{L}_{\text{reg}}(\mathbf{W}_{\text{inv}}) + \mathcal{L}_{\text{tg}}$ | 0.045 | 0.067 | 0.063 | 0.046 | 0.050 | 0.048 | 0.263 | 1.270 | 0.081 | 0.039 | 0.079 | 0.042 | 5.167 |
| $\mathcal{L}_{\text{reg}}(\mathbf{W}_{\text{inv}}) + \mathcal{L}_{\text{F}}$ | 0.042 | 0.065 | 0.061 | 0.044 | 0.050 | 0.045 | 0.245 | 1.042 | 0.078 | 0.037 | 0.050 | 0.041 | 2.333 |
| $\mathcal{L}_{\text{reg}}(\mathbf{W}_{\text{inv}}) + \mathcal{L}_{\text{F}} + \mathcal{L}_{\text{S}}$ | 0.043 | 0.068 | 0.061 | 0.047 | 0.055 | 0.044 | 0.255 | 0.901 | 0.080 | 0.037 | 0.036 | 0.040 | 3.083 |
| (Our loss) $\mathcal{L}_{\text{reg}}(\mathbf{W}_{\text{inv}}) + \mathcal{L}_{\text{F}} + \mathcal{L}_{\text{consis}}$ | 0.043 | 0.065 | 0.061 | 0.042 | 0.045 | 0.045 | 0.249 | 0.919 | 0.077 | 0.037 | 0.041 | 0.041 | **1.917** |
| w/o High Resolution | 0.043 | 0.065 | 0.061 | 0.042 | 0.045 | 0.045 | 0.249 | 0.919 | 0.077 | 0.037 | 0.041 | 0.041 | 1.417 |
| (Ours) w/ Efficient High Resolution | 0.043 | 0.068 | 0.061 | 0.038 | 0.042 | 0.046 | 0.238 | 0.964 | 0.080 | 0.031 | 0.035 | 0.041 | **1.333** |
| w/ VGGT High Resolution | 0.056 | 0.081 | 0.067 | 0.057 | 0.036 | 0.045 | 0.237 | 0.355 | 0.064 | 0.039 | 0.060 | 0.050 | 1.750 |
| (Ours) w/ Efficient High Resolution | 0.058 | 0.059 | 0.061 | 0.037 | 0.029 | 0.046 | 0.235 | 0.289 | 0.071 | 0.034 | 0.024 | 0.042 | **1.250** |

**Training Objective.** In this subsection, we conduct the ablation study on the "Data3" training data composition. Starting from the original VGGT loss, we ablate the effectiveness of $\mathcal{L}_{\text{sg}}$ and $\mathcal{L}_{\text{conf}}$, and present the results in Table 1. Unexpectedly, when we remove the $\mathcal{L}_{\text{sg}}$ and $\mathcal{L}_{\text{conf}}$, the overall performance improves, which shows the harm of the spatial gradient loss and the confidence map weight. In contrast, we follow MoGe Wang et al. (2025c) to use the inverse of the depth values as a fixed weight map leads to further performance improvement ("$\mathcal{L}_{\text{reg}} + \mathcal{L}_{\text{conf}}$" vs. "$\mathcal{L}_{\text{reg}}(\mathbf{W}_{\text{inv}})$"). We attribute this to the possibility that the learned loss weight may cause the model to avoid difficult samples, whereas the reciprocal of the depth values serves as a natural measure of confidence.

> **Finding 2.** For the original loss of VGGT, the spatial gradient loss and the learnable confidence-aware weighting strategy unexpectedly lead to performance degradation. Employing the inverse of depth values as a weight map proves to be a more effective choice.

Based on $\mathcal{L}_{\text{reg}}(\mathbf{W}_{\text{inv}})$, we extend our exploration to more loss functions. Similar to $\mathcal{L}_{\text{sg}}$, we observe that the temporal gradient loss $\mathcal{L}_{\text{tg}}$ Chen et al. (2025) also negatively impacts performance.

$$\mathcal{L}_{\text{tg}}(\hat{\boldsymbol{\xi}}, \boldsymbol{\xi}, \mathbf{W}) = \mathbb{E}_{t \in \mathcal{T}, p \in \mathcal{M}} \left\| \mathbf{W}_{t,p} \cdot (\nabla_t \hat{\boldsymbol{\xi}}_{t,p} - \nabla_t \boldsymbol{\xi}_{t,p}) \right\|, \tag{6}$$

Table 2: The training data components for ablation study. From "Data1" to "Data3", the data volume increases progressively. While "Data1" and "Data2" contain only high-quality data, "Data3" incorporates additional noisy data.

| Name | Training Data Components |
|------|------------------------|
| Data1 | Hypersim, ScanNet++, Virtual KITTI2, MVS-Synth, Spring, UnrealStereo4K |
| Data2 | "Data1", Tartanair, Parallel Domain, TartanGround |
| Data3 | "Data2", ScanNet, ARKitScenes, GraspNet, BlendedMVS |

Table 3: The parameter count (Params.) and frames per second (FPS) of VGGT and CARVE tested on a single NVIDIA H200 GPU. The number of parameters is reported in millions, and the FPS is averaged over 100 runs following 10 preliminary warm-up runs.

| Method | Params. (M) | Image Resolution | FPS |
|--------|-------------|------------------|-----|
| VGGT | 1189.01 | $518 \times 518$ | 24.85 |
| | | $1036 \times 1036$ | 2.54 |
| CARVE (Ours) | 1214.21 | $1036 \times 1036$ | 15.26 |

Table 4: Efficiency metrics for different input frames and resolutions on a single NVIDIA H200 GPU. The number of floating point operations is measured in teraFLOPs (TFLOPs), and Memory is measured in gibibytes (GiB). "(H$\times$ W)" represents the input image resolution.

| | VGGT ($518 \times 518$) | | VGGT ($1036 \times 1036$) | | CARVE (Ours, $1036 \times 1036$) | |
|---|---|---|---|---|---|---|
| # Frames | TFLOPs | Peek GPU Mem (GiB) | TFLOPs | Peek GPU Mem (GiB) | TFLOPs | Peek GPU Mem (GiB) |
| 8 | 25.57 | 8.81 | 101.99 | 21.80 | 52.97 | 9.08 |
| 16 | 51.14 | 10.99 | 203.98 | 30.49 | 105.93 | 11.40 |
| 32 | 102.28 | 15.36 | 407.97 | 47.89 | 211.87 | 16.05 |
| 64 | 204.56 | 25.61 | 815.93 | 88.81 | 423.73 | 25.71 |
| 128 | 409.13 | 46.75 | OOM | OOM | 847.47 | 46.86 |
| 256 | 818.25 | 89.02 | OOM | OOM | 1694.94 | 89.14 |

where $\nabla_t$ represents the difference between temporally nearby pixels. For the alignment strategy of the training loss, rather than applying a global scale to align the entire sequence with the ground truth, MoGe Wang et al. (2025c) proposes to apply separate scale-shift alignment for each frame ($L_F$) and each sampled local 3D spherical region of the point cloud ($L_S$).

$$\mathcal{L}_F(\hat{\boldsymbol{\xi}}, \boldsymbol{\xi}, \mathbf{W}) = \mathbb{E}_{t \in \mathcal{T}, p \in \mathcal{M}} \left\| \mathbf{W}_{t,p} \cdot (\mathbf{a}_t \cdot \hat{\boldsymbol{\xi}}_{t,p} + \mathbf{B}_t - \boldsymbol{\xi}_{t,p}) \right\|,$$

$$\mathcal{L}_S(\hat{\boldsymbol{\xi}}, \boldsymbol{\xi}, \mathbf{W}) = \mathbb{E}_{\mathcal{S}_j \in \mathcal{S}, p \in \mathcal{S}_j} \left\| \mathbf{W}_{j,p} \cdot (\mathbf{a}_j \cdot \hat{\boldsymbol{\xi}}_{j,p} + \mathbf{B}_j - \boldsymbol{\xi}_{j,p}) \right\|, \quad (7)$$

$$\mathcal{S}_j = \{p \mid \|\mathbf{P}_p - \mathbf{P}_j\| \le r_j, p \in \mathcal{M}\},$$

where $\mathbf{a}_t$ and $\mathbf{B}_t$ are the scale and shift alignment parameters for each frame $t$, and $\mathbf{a}_j$ and $\mathbf{B}_j$ are the parameters for each sampled local 3D spherical region $\mathcal{S}_j$, with 3D region radius $r_j$. All these scale-shift parameters are computed with ROE alignment Wang et al. (2025c). As presented in Table 1, we observe that supervising with both per-sequence and per-frame alignment improves performance ("$\mathcal{L}_{reg}(\mathbf{W}_{inv})$" vs. "$\mathcal{L}_{reg}(\mathbf{W}_{inv}) + \mathcal{L}_F$"), while the local region alignment unexpectedly results in a decrease in performance ("$\mathcal{L}_{reg}(\mathbf{W}_{inv}) + \mathcal{L}_F$" vs. "$\mathcal{L}_{reg}(\mathbf{W}_{inv}) + \mathcal{L}_F + \mathcal{L}_S$").

Furthermore, we propose a novel consistency loss $\mathcal{L}_{consis}$ to enforce the consistency between the estimated point map and the unprojected one.

$$\mathcal{L}_{consis}(\hat{\mathbf{P}}, \hat{\mathbf{D}}, \hat{\mathbf{r}}, \hat{\mathbf{t}}, \hat{\boldsymbol{\theta}}) = \mathbb{E}_{p \in \mathcal{M}} \left| \hat{\mathbf{P}}_{unproj}(p) - \hat{\mathbf{P}}(p) \right|,$$

$$\hat{\mathbf{P}}_{unproj}(p) = \hat{\mathbf{R}}(\hat{\mathbf{D}}(p)\hat{\mathbf{K}}^{-1}p) + \hat{\mathbf{t}}, \ \hat{\mathbf{R}} = \mathcal{H}(\hat{\mathbf{r}}),$$

$$\hat{\mathbf{K}} = \text{Intrinsics}(\hat{f}_x, \hat{f}_y, \hat{c}_x, \hat{c}_y), \ \hat{c}_x = W/2, \hat{c}_y = H/2 \quad (8)$$

$$\hat{f}_x = \frac{W}{2 \tan\left(\hat{\boldsymbol{\theta}}_x/2\right)}, \ \hat{f}_y = \frac{H}{2 \tan\left(\hat{\boldsymbol{\theta}}_y/2\right)},$$

where $\text{Intrinsics}(\cdot, \cdot, \cdot, \cdot)$ computes $3 \times 3$ camera intrinsic matrix from the focal length and the optical center, and $\mathcal{H}(\cdot)$ transforms a rotation quaternion to a $3 \times 3$ camera rotation matrix. As shown in Table 1, the comparison between "$\mathcal{L}_{reg}(\mathbf{W}_{inv}) + \mathcal{L}_F$" and "$\mathcal{L}_{reg}(\mathbf{W}_{inv}) + \mathcal{L}_F + \mathcal{L}_{consis}$" demonstrates that enforcing consistency can lead to improved robustness and accuracy.

> ***Finding 3.*** 1) Similarly, the temporal gradient loss leads to a reduction in performance. 2) Supervision using both per-sequence and per-frame alignment enhances results, whereas the local region alignment unexpectedly brings a reduction. 3) Enforcing consistency between the estimated point cloud and the unprojected one yields improvement.

Figure 1: Network architecture of our proposed CARVE model. We extract the high-resolution feature and fuse it into the low-resolution main branch with frame-wise cross attention modules and zero-initialized residual gate parameters $\beta$.

**Efficient High-Resolution Adaptation.** It is well recognized that higher-resolution inputs typically enhance the performance of computer vision tasks. However, for the attention module of the VGGT transformer block, directly upsampling the input image by a factor of 2 theoretically results in $4\times$ tokens and $16\times$ computational complexity. In practice, we report TFLOPs, GPU memory usage, and FPS of VGGT under both low- and high-resolution input settings in Table 4 and Table 3. Despite the adoption of several engineering optimizations (see the appendix for details), high-resolution input still results in $4\times$ TFLOPs, $3\times$ to $4\times$ GPU memory usage, and $0.1\times$ FPS.

In contrast, we propose an efficient high-resolution adaptation network as illustrated in Figure 1. We extract the high-resolution feature and fuse it to the low-resolution main branch before sending it to the transformer block with frame-wise cross attention modules. The low-resolution image serves as the query, while the high-resolution image serves as the key and value. Similar to VGGT's frame-wise attention, the cross-attention is computed between low- and high-resolution image pairs of the same frame. To prevent the pretrained parameters from being degraded, inspired by ResNet He et al. (2016), we treat the cross-attention outputs as a residual branch, which is added to the main branch for each cross-attention block after being scaled by a learnable gating parameter. These gating parameters are initialized to zero. For the depth head and point head, we simply upsample the feature prior to the last few convolution layers. The formulation is as follows.

$$\hat{\mathbf{f}}_{\text{img\_low}} = \text{Encoder}(\mathbf{I}_{\text{low}}), \ \hat{\mathbf{f}}_{\text{img\_high}} = \text{Encoder}(\mathbf{I}_{\text{high}}),$$
$$\hat{\mathbf{f}}_{\text{img}} = \hat{\mathbf{f}}_{\text{img\_low}} + \beta \cdot \text{CrossAttn}(\hat{\mathbf{f}}_{\text{img\_low}}, \hat{\mathbf{f}}_{\text{img\_high}}), \quad (9)$$
$$(\hat{\mathbf{f}}_{\text{geo}}, \hat{\mathbf{f}}_{\text{cam}}) = \text{Transformer}(\hat{\mathbf{f}}_{\text{img}}, \mathbf{f}_{\text{cam\_init}}),$$

where the feature $\hat{\mathbf{f}}_{\text{img\_low}}$ and $\hat{\mathbf{f}}_{\text{img\_high}}$ are extracted separately from the low-resolution image $\mathbf{I}_{\text{low}}$ and high-resolution image $\mathbf{I}_{\text{high}}$, and $\beta$ is the learnable gate parameter with zero initialization. The cross-attention block $\text{CrossAttn}(\cdot, \cdot)$ takes $\hat{\mathbf{f}}_{\text{img\_low}}$ as query and $\hat{\mathbf{f}}_{\text{img\_high}}$ as key and value. The fused feature $\hat{\mathbf{f}}_{\text{img}}$ shares the same dimensionality as $\hat{\mathbf{f}}_{\text{img\_low}}$, allowing it to seamlessly replace the original low-resolution feature in subsequent modules. As demonstrated in Table 1, our architecture enhances the overall performance ("w/o High Resolution" vs. "w/ Efficient High Resolution"). Furthermore, our proposed efficient high-resolution architecture even outperforms the direct input upsampling strategy ("w/ Efficient High Resolution" vs. "w/ VGGT High Resolution" in gray color, the evaluations are conducted with a maximum of 100 frames due to GPU memory constraints.). We hypothesize that this improvement arises from two factors: 1) Our efficient architecture processes both high- and low-resolution images, where the integration of multi-resolution features proves beneficial. 2) High-resolution inputs may conflict with the original VGGT pretrained weights, which were learned from low-resolution data. For efficiency metrics, our proposed high-resolution architecture achieves substantial computational efficiency, requiring only $0.3\times$ to $0.4\times$ GPU memory, $0.5\times$ TFLOPs, and delivering up to $6\times$ higher FPS during inference, as reported in Table 3.

> ***Finding* 4.** Leveraging an efficient high-resolution adaptation architecture, the network demonstrates a superior balance between performance and computational efficiency.

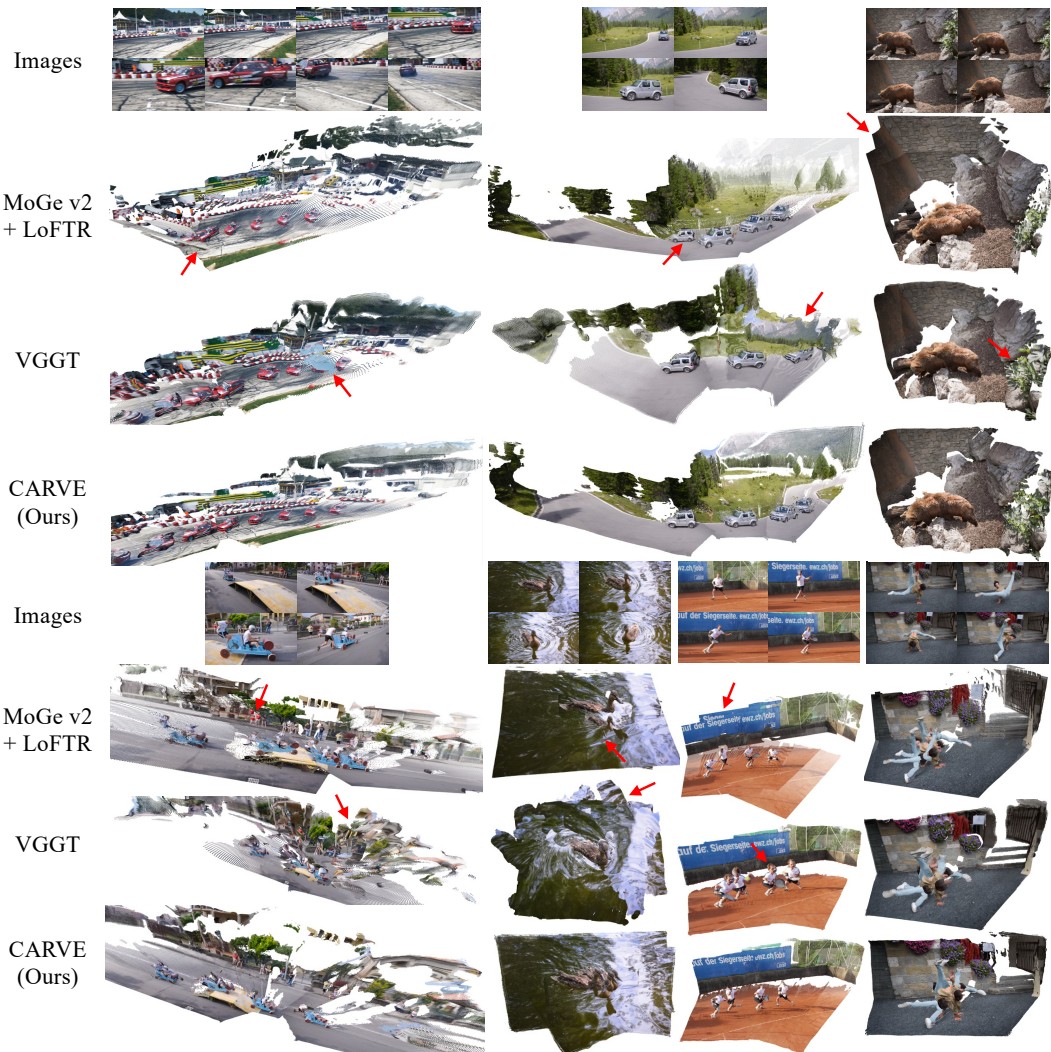

Figure 2: Qualitative results of point cloud estimation on in-the-wild images. The red arrows highlight instances of failed estimations, including incorrect camera pose estimation, abnormal geometry scaling, and inconsistencies between frames.

Table 5: Quantitative results of point cloud estimation on KITTI, 7-Scenes, and TUM.

| Method | KITTI | | | | 7-Scenes | | | | TUM | | | | Rank↓ |
|---|---|---|---|---|---|---|---|---|---|---|---|---|---|
| | C-L1↓ | F@5↑ | F@25↑ | F@50↑ | C-L1↓ | F@5↑ | F@25↑ | F@50↑ | C-L1↓ | F@5↑ | F@25↑ | F@50↑ | |
| MoGe v2 + LoFTR | 0.726 | 0.142 | 0.562 | 0.750 | 0.161 | 0.242 | 0.777 | 0.950 | 0.221 | 0.199 | 0.696 | 0.844 | 4.4 |
| Spann3R | 2.359 | 0.044 | 0.296 | 0.452 | 0.101 | 0.375 | 0.922 | 0.987 | 0.122 | 0.498 | 0.860 | 0.949 | 4.0 |
| CUT3R | 0.708 | 0.125 | 0.491 | 0.666 | 0.126 | 0.328 | 0.849 | 0.982 | 0.140 | 0.317 | 0.851 | 0.965 | 3.8 |
| Fast3R | 4.974 | 0.088 | 0.357 | 0.501 | 0.655 | 0.045 | 0.226 | 0.422 | 0.936 | 0.028 | 0.153 | 0.261 | 5.8 |
| VGGT | 0.296 | 0.220 | 0.688 | 0.842 | 0.049 | 0.660 | **0.988** | 0.997 | 0.051 | 0.712 | 0.980 | 0.993 | 1.9 |
| CARVE (Ours) | **0.238** | **0.257** | **0.767** | **0.892** | **0.043** | **0.720** | 0.986 | **0.998** | **0.029** | **0.861** | **0.991** | **0.997** | **1.1** |

Table 6: Quantitative results of point cloud estimation on HAMMER, Bonn, and ETH3D.

| Method | HAMMER | | | | Bonn | | | | ETH3D | | | | Rank↓ |
|---|---|---|---|---|---|---|---|---|---|---|---|---|---|
| | C-L1↓ | F@5↑ | F@25↑ | F@50↑ | C-L1↓ | F@5↑ | F@25↑ | F@50↑ | C-L1↓ | F@5↑ | F@25↑ | F@50↑ | |
| MoGe v2 + LoFTR | 0.030 | 0.872 | **1.000** | **1.000** | 0.174 | 0.226 | 0.765 | 0.941 | 1.889 | 0.002 | 0.026 | 0.066 | 3.7 |
| Spann3R | 0.041 | 0.727 | **1.000** | **1.000** | 0.114 | 0.354 | 0.894 | 0.978 | 2.479 | 0.067 | 0.216 | 0.366 | 3.2 |
| CUT3R | 0.049 | 0.599 | **1.000** | **1.000** | 0.133 | 0.345 | 0.846 | 0.966 | 2.579 | 0.089 | 0.264 | 0.397 | 3.5 |
| Fast3R | 0.062 | 0.488 | **1.000** | **1.000** | 0.983 | 0.019 | 0.134 | 0.285 | 3.901 | 0.000 | 0.000 | 0.000 | 5.2 |
| VGGT | 0.035 | 0.828 | 0.999 | **1.000** | 0.057 | 0.645 | 0.972 | 0.987 | **0.202** | 0.410 | **0.787** | **0.915** | 2.2 |
| CARVE (Ours) | **0.012** | **0.999** | **1.000** | **1.000** | **0.043** | **0.720** | **0.986** | **0.998** | 0.236 | **0.423** | 0.765 | 0.867 | **1.2** |

Table 7: Quantitative results of video depth estimation on seven datasets.

| Method | KITTI Rel↓ | δ↑ | 7-Scenes Rel↓ | δ↑ | TUM Rel↓ | δ↑ | HO3D Rel↓ | δ↑ | HAMMER Rel↓ | δ↑ | Bonn Rel↓ | δ↑ | ETH3D Rel↓ | δ↑ | Rank↓ |
|---|---|---|---|---|---|---|---|---|---|---|---|---|---|---|---|
| MoGe v2 + LoFTR | 0.453 | 0.430 | 0.217 | 0.675 | 0.225 | 0.593 | 0.278 | 0.811 | 0.036 | **0.997** | 0.171 | 0.797 | 0.242 | 0.690 | 3.6 |
| CUT3R | 0.134 | 0.835 | 0.095 | 0.909 | 0.101 | 0.909 | 0.366 | 0.689 | 0.054 | 0.972 | 0.063 | 0.948 | 0.111 | 0.884 | 3.2 |
| Fast3R | 0.254 | 0.645 | 0.312 | 0.476 | 0.377 | 0.446 | 0.524 | 0.587 | 0.135 | 0.838 | 0.339 | 0.551 | 0.568 | 0.335 | 4.9 |
| VGGT | 0.094 | 0.917 | 0.069 | 0.930 | 0.062 | 0.954 | 0.270 | 0.755 | 0.046 | 0.968 | 0.054 | 0.953 | 0.043 | 0.978 | 2.3 |
| CARVE (Ours) | **0.082** | **0.933** | **0.062** | **0.940** | **0.040** | **0.976** | **0.220** | **0.869** | **0.020** | 0.996 | **0.041** | **0.959** | **0.023** | **0.997** | **1.1** |

Table 8: Quantitative evaluation of camera pose and intrinsic on KITTI, 7-Scenes, TUM, and HO3D.

| Method | KITTI FoV Rel↓ | ATE↓ | RPE-R↓ | RPE-T↓ | 7-Scenes FoV Rel↓ | ATE↓ | RPE-R↓ | RPE-T↓ | TUM FoV Rel↓ | ATE↓ | RPE-R↓ | RPE-T↓ | HO3D FoV Rel↓ | Rank↓ |
|---|---|---|---|---|---|---|---|---|---|---|---|---|---|---|
| MoGe v2 | 0.162 | – | – | – | 0.192 | – | – | – | 0.124 | – | – | – | 0.067 | 4.5 |
| CUT3R | 0.088 | 3.521 | 0.110 | 9.542 | 0.089 | 0.305 | 0.158 | 0.453 | 0.071 | 0.298 | 0.226 | 0.472 | 0.120 | 3.4 |
| Fast3R | 0.079 | 106.082 | 0.161 | 125.443 | 0.075 | 1.696 | 1.056 | 2.576 | 0.028 | 1.189 | 1.257 | 1.897 | **0.012** | 3.3 |
| VGGT | 0.084 | 1.113 | **0.015** | 2.177 | 0.076 | 0.073 | **0.062** | 0.117 | **0.020** | 0.047 | 0.038 | 0.063 | 0.109 | 2.1 |
| CARVE (Ours) | **0.078** | **0.664** | 0.016 | **1.740** | **0.024** | **0.052** | 0.064 | **0.104** | 0.049 | **0.041** | **0.032** | **0.060** | 0.039 | **1.4** |

Table 9: Quantitative evaluation of camera pose and intrinsic on HAMMER, Bonn, and ETH3D.

| Method | HAMMER FoV Rel↓ | ATE↓ | RPE-R↓ | RPE-T↓ | Bonn FoV Rel↓ | ATE↓ | RPE-R↓ | RPE-T↓ | ETH3D FoV Rel↓ | ATE↓ | RPE-R↓ | RPE-T↓ | Rank↓ |
|---|---|---|---|---|---|---|---|---|---|---|---|---|---|
| MoGe v2 | 0.084 | – | – | – | 0.136 | – | – | – | 0.058 | – | – | – | 4.7 |
| CUT3R | 0.063 | 0.010 | 0.023 | 0.018 | 0.029 | 0.247 | 0.129 | 0.375 | 0.032 | 2.859 | 0.364 | 3.251 | 3.1 |
| Fast3R | 0.062 | 0.119 | 0.166 | 0.187 | **0.025** | 0.669 | 0.722 | 1.089 | 0.075 | 13.074 | 1.491 | 16.466 | 3.8 |
| VGGT | 0.040 | **0.001** | **0.003** | **0.002** | 0.040 | 0.075 | 0.042 | 0.091 | 0.020 | 1.804 | **0.021** | 2.143 | 1.8 |
| CARVE (Ours) | **0.035** | **0.001** | 0.004 | 0.003 | 0.028 | **0.044** | **0.029** | **0.056** | **0.018** | **0.184** | 0.022 | **0.223** | **1.3** |

Table 10: Quantitative results of monocular depth estimation on seven datasets.

| Method | KITTI Rel↓ | δ↑ | 7-Scenes Rel↓ | δ↑ | TUM Rel↓ | δ↑ | HO3D Rel↓ | δ↑ | HAMMER Rel↓ | δ↑ | Bonn Rel↓ | δ↑ | ETH3D Rel↓ | δ↑ | Rank↓ |
|---|---|---|---|---|---|---|---|---|---|---|---|---|---|---|---|
| MoGe | **0.094** | 0.904 | 0.070 | 0.938 | 0.055 | 0.966 | 0.282 | 0.788 | 0.028 | 0.988 | **0.034** | **0.986** | 0.035 | **0.988** | 2.0 |
| MoGe v2 | 0.098 | **0.908** | 0.077 | 0.932 | 0.057 | 0.964 | 0.256 | 0.837 | **0.023** | **0.996** | 0.037 | 0.984 | 0.036 | 0.986 | 2.6 |
| CUT3R | 0.129 | 0.849 | 0.073 | 0.933 | 0.065 | 0.942 | 0.379 | 0.703 | 0.049 | 0.974 | 0.042 | 0.975 | 0.061 | 0.952 | 4.6 |
| Fast3R | 0.274 | 0.594 | 0.247 | 0.591 | 0.285 | 0.597 | 0.550 | 0.556 | 0.143 | 0.813 | 0.230 | 0.646 | 0.404 | 0.527 | 6.0 |
| VGGT | 0.125 | 0.855 | 0.070 | 0.934 | 0.062 | 0.948 | 0.269 | 0.775 | 0.054 | 0.972 | 0.042 | 0.975 | 0.043 | 0.974 | 3.9 |
| CARVE (Ours) | 0.106 | 0.885 | **0.066** | **0.941** | **0.049** | **0.969** | **0.236** | 0.851 | 0.028 | 0.994 | 0.035 | 0.985 | **0.033** | 0.985 | **1.7** |

# 4 EXPERIMENTS

We scale up the training process with the training objects of "$\mathcal{L}_{\text{reg}}(\mathbf{W}_{\text{inv}}) + \mathcal{L}_{\text{F}} + \mathcal{L}_{\text{consis}}$", training data of "Data3", and our proposed efficient high-resolution architecture. The model is initialized with the VGGT Wang et al. (2025a) pretrained weights for common parameters. More training and evaluation details are provided in the appendix. We evaluate visual geometry estimation across multiple datasets, including KITTI Geiger et al. (2013), 7-Scenes Shotton et al. (2013), HO3D Hampali et al. (2020), TUM Sturm et al. (2012), ETH3D Schops et al. (2017), HAMMER Jung et al. (2023), and Bonn Palazzolo et al. (2019).

## 4.1 POINT CLOUD ESTIMATION

For point cloud evaluation, we align the stacked point cloud with the corresponding stacked ground-truth one using a similarity transformation comprising a scale factor, a rotation matrix, and a translation vector. We report the Chamfer L1 distance (C-L1) and the F-score at thresholds of 5cm (F@5), 25cm (F@25), and 50cm (F@50). The estimated and ground-truth point clouds are downsampled using a voxel size of 2cm for fast evaluation.

We compare with the monocular reconstruction model MoGe v2 Wang et al. (2025d) ("MoGe v2 + LoFTR"), multi-view reconstruction model Spann3R Wang & Agapito (2025), CUT3RWang et al. (2025b), Fast3RYang et al. (2025) and VGGT Wang et al. (2025a). We use LoFTR Sun et al. (2021) for feature extraction and matching, and compute the similarity transformation between frames using the matched points to achieve alignment of the results from MoGe v2 Wang et al. (2025d).

Quantitative comparisons are shown in Table 5 and Table 6. As observed, "MoGe v2 + LoFTR" performs well on datasets with limited viewpoint variation, such as HAMMER. However, its reliance on accurate matching information limits its effectiveness in more complex scenarios. VGGT outperforms other multi-view methods, including Spann3R, CUT3R, and Fast3R, which can be attributed to its temporally scalable network framework that improves its ability to model long-range dependencies across views. Our CARVE, benefiting from our comprehensive analysis and improvements, achieves state-of-the-art robustness and accuracy across six evaluation datasets.

Qualitative comparisons are presented in Figure 2. MoGe v2 + LoFTR demonstrates detailed visualizations but suffers from poor temporal consistency across frames. In contrast, VGGT produces consistent results, but its accuracy is relatively suboptimal. Our proposed method, CARVE, effectively balances both spatial accuracy and temporal consistency, achieving superior overall performance.

### 4.2 VIDEO DEPTH ESTIMATION

We align the predicted video maps with the ground-truth video maps through only a global scale value. We report the absolute relative error Rel$= \mathbb{E}_{p\in\mathcal{M}} |(\hat{\mathbf{D}}_p - \mathbf{D}_p)|/\mathbf{D}_p$ and the percentage of pixels $\delta = \mathbb{E}_{p\in\mathcal{M}} \max(\hat{\mathbf{D}}_p/\mathbf{D}_p, \mathbf{D}_p/\hat{\mathbf{D}}_p) < 1.25$. Spann3R is not evaluated due to the pure world-coordinates point clouds output. For MoGe v2, we align the depth with LoFTR for each frame. A single scale factor is applied to the entire image sequence to align the video depth sequence.

Quantitative comparisons of video depth estimation are shown in Table 7. MoGe v2 is limited by the inaccurate matching information. Similarly, VGGT outperforms other multi-view methods, including Spann3R, CUT3R, and Fast3R. Our proposed CARVE achieves state-of-the-art performance.

### 4.3 CAMERA POSE AND INTRINSICS ESTIMATION

For camera pose estimation, we follow Sturm et al. (2012) to align the predicted camera pose with the ground truth and evaluate the absolute trajectory error (ATE), relative pose error of rotation (RPE-R), and translation (RPE-T). For camera intrinsics, we evaluate the accuracy with the "FOV Rel", which is defined as the absolute relative error of the field of view (FOV Rel$= \mathbb{E}_t |\hat{\boldsymbol{\theta}}_t - \boldsymbol{\theta}_t|/\boldsymbol{\theta}_t$) to ensure the evaluation of camera intrinsics is independent of image resolution. For MoGe v2, we only evaluate the FoV Rel metric.

Quantitative comparisons are shown in Table 8 and Table 9. The results show that our proposed CARVE consistently outperforms other approaches on six evaluation datasets.

### 4.4 MONOCULAR DEPTH ESTIMATION

Similar to video depth evaluation, we compare the monocular depth estimation metrics with other feed-forward reconstruction methods. We continue to use the absolute relative error (Rel.) and the threshold accuracy ($\delta < 1.25$, denoted as $\delta$) for monocular depth estimation, but unlike video depth estimation, we perform per-image alignment. Quantitative comparisons are shown in Table 10. MoGe exhibits strong performance and demonstrates notable competitiveness in monocular depth estimation tasks. Remarkably, despite not being explicitly optimized for monocular depth estimation, our proposed CARVE achieves competitive performance, which benefits from the VGGT architecture and shows the robustness of CARVE.

## 5 CONCLUSION

In this work, we analyze the core components necessary for visual geometry estimation. We begin with a systematic study of how training data and objective design affect performance. Then, we introduce a novel consistency loss and a lightweight, effective feature-fusion module that enables accurate inference from high-resolution inputs without incurring prohibitive computational cost. These contributions culminate in CARVE, a unified framework that achieves state-of-the-art performance across diverse benchmark datasets.

## REPRODUCIBILITY STATEMENT

To ensure the reproducibility of our research, we provide extensive details throughout the paper and its appendix. The necessary theoretical background and foundational concepts are discussed in detail in Section 2 and 3. All datasets used in the experiments are publicly available and are described in detail with references in the Appendix B. We have also made every effort to document the implementation details, including the model architecture, training protocol, evaluation metrics, and hyperparameter settings, which are described in Section 4 and further elaborated in the Appendix B. Upon publication, we will release the source code in a public repository.

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

APPENDIX

## A  USE OF LLMS

In this work, Large Language Models (LLMs) were used to assist in refining and polishing the language of the manuscript. The LLM was employed primarily for tasks such as improving clarity, enhancing the flow of text, correcting grammatical errors, and rephrasing sections of the paper to make the writing more concise and readable. The model was not involved in the conceptualization or ideation of the research or in the analysis of results. All scientific content, including methodologies, experiments, and findings, were independently generated by the authors.

## B  EXPERIMENTAL SETTING DETAILS

In the appendix, we focus on providing additional details and quantitative results. 1) We present more training and evaluation details for the ablation study and main experiments; 2) We include extended visualization results in Figure 3.

**Common Training Details.** The experiments were conducted on a server running Ubuntu 22.04 equipped with two Intel Xeon Platinum 8558 CPUs (192 threads in total) and 1.8 TB system memory. The system was configured with eight NVIDIA H200 GPUs using NVIDIA driver 570.133.20 and CUDA 12.4.

The model is initialized with the VGGT Wang et al. (2025a) pretrained weights for commom parameters. Unless otherwise specified, we freeze the ViT feature extractor of VGGT Wang et al. (2025a) and train the remaining components. The regression loss of the camera head is scaled by a factor of 5 to balance between tasks .Training is performed using the AdamW optimizer Loshchilov & Hutter (2019) with $\beta_1 = 0.9$, $\beta_2 = 0.99$, and a weight decay of 0.01. The learning rate is scheduled using the OneCycleLR policy Smith & Topin (2019). The longer side of the low-resolution input image is resized to 518 pixels, and the shorter side is then randomly cropped to one of (448, 378, 308, 238) pixels. For data augmentation, we use random Gaussian blur, Gaussian noise, color jittering, and grayscale. The predicted point cloud, depth map, and camera translation are aligned with the ground-truth values via a scale factor for each sequence before computing the training loss. For the training datasets, they are categorized into three groups: indoor scenes, autonomous driving, and others. To ensure balanced dataset components, we normalize the dataset sizes such that each group contributes an equal volume of data, and individual datasets in each group are expanded to maintain intra-group balance. To accelerate training and reduce CUDA memory requirement, we employ Flash Attention v2 Dao (2024) in all attention blocks and utilize ZeRO Stage 2 optimization provided by the HuggingFace Accelerate framework. We use PyTorch's gradient checkpointing technique to reduce the CUDA memory usage. A random seed of 2025 is used in our experiment.

**Training Details for Ablation Study.** The model is trained with a learning rate of 3e-6 for 30K iterations on a single NVIDIA H200 GPU, and we employ a dynamic batch size with the sequence length varying between 2 and 24 frames. We constrain the total number of input images to a maximum of 24 for each iteration.

For the training data ablation study, we adopt the original vggt loss component $\mathcal{L}_{\text{reg}} + \mathcal{L}_{\text{msg}} + \mathcal{L}_{\text{conf}}$, weighted by the inverse value of ground-truth depth. For the training objective ablation, we use the training dataset component of "Data3", including ScanNet++ Yeshwanth et al. (2023), Hypersim Roberts et al. (2021), ScanNet Dai et al. (2017), ARKitScenes Baruch et al. (2021), GraspNet Fang et al. (2020), Virtual KITTI2 Cabon et al. (2020), MVS-Synth Huang et al. (2018b), Parallel Domain Thomas et al. (2021), Spring Mehl et al. (2023), UnrealStereo4K Tosi et al. (2021), Tartanair Wang et al. (2020), TartanGround Patel et al. (2025), and BlendedMVS Yao et al. (2020).

**Training Details for CARVE.** Based on the preceding analysis, we follow the VGGT Wang et al. (2025a) model with approximately 1.2 billion parameters in total, and adopt our efficient high-resolution adaptation with two cross-attention blocks to handle the input high-resolution image.

During training, the corresponding high-resolution image maintains twice the resolution of the low-resolution input. For each GPU, we employ a dynamic batch size with the sequence length varying between 2 and 50 frames, and constrain the total number of input images to a maximum of 50 for

| Training Dataset | Data Type | Data Quality | Sequences | Images |
|---|---|---|---|---|
| ScanNet++ Yeshwanth et al. (2023) | Indoor | High | 280 | 175661 |
| Hypersim Roberts et al. (2021) | Indoor | High | 743 | 72019 |
| ScanNet Dai et al. (2017) | Indoor | Middle | 1513 | 2477378 |
| ARKitScenes Baruch et al. (2021) | Indoor | Middle | 2312 | 2049625 |
| GraspNet Fang et al. (2020) | Indoor | Middle | 380 | 97280 |
| Virtual KITTI2 Cabon et al. (2020) | Driving | High | 100 | 42520 |
| MVS-Synth Huang et al. (2018b) | Driving | High | 120 | 12000 |
| Parallel Domain Thomas et al. (2021) | Driving | High | 367 | 347480 |
| Spring Mehl et al. (2023) | Other | High | 74 | 10000 |
| UnrealStereo4K Tosi et al. (2021) | Other | High | 18 | 16400 |
| Tartanair Wang et al. (2020) | Other | High | 738 | 613274 |
| TartanGround Patel et al. (2025) | Other | High | 14 | 18484 |
| BlendedMVS Yao et al. (2020) | Other | Middle | 615 | 132961 |
| Total | – | – | 7274 | 6065082 |

Table 11: The data type, quality, sequence count and image count of training datasets.

| Eval Dataset | Data Type | Sequences | Avg. Frames | Stride |
|---|---|---|---|---|
| KITTI Geiger et al. (2013) | Driving | 6 | 107.8 | 1 |
| 7-Scenes Shotton et al. (2013) | Indoor | 46 | 187.0 | 5 |
| HO3D Hampali et al. (2020) | Indoor & Object | 13 | 198.5 | 5 |
| TUM Sturm et al. (2012) | Indoor | 9 | 199 | 3 |
| ETH3D Schops et al. (2017) | Indoor & Outdoor | 11 | 34.5 | 1 |
| HAMMER Jung et al. (2023) | Indoor & Object | 9 | 130.0 | 1 |
| Bonn Palazzolo et al. (2019) | Indoor | 26 | 185.7 | 3 |

Table 12: The data type, sequence count, average frames per sequence of evaluation datasets. "Stride" means that we sample every "Stride" element from the sequence.

each iteration. We use a learning rate of 1e-5, and train the model for 30K iterations on 8 NVIDIA H200 GPUs. For training loss, we adopt the final loss plan of "$\mathcal{L}_{\text{reg}}(\mathbf{W}_{\text{inv}}) + \mathcal{L}_{\text{F}} + \mathcal{L}_{\text{consis}}$".

For training data, we leverage a diverse set of datasets same to "Data4", including ScanNet++ Yesh-wanth et al. (2023), Hypersim Roberts et al. (2021), ScanNet Dai et al. (2017), ARKitScenes Baruch et al. (2021), GraspNet Fang et al. (2020), Virtual KITTI2 Cabon et al. (2020), MVS-Synth Huang et al. (2018b), Parallel Domain Thomas et al. (2021), Spring Mehl et al. (2023), UnrealStereo4K Tosi et al. (2021), Tartanair Wang et al. (2020), TartanGround Patel et al. (2025), and BlendedMVS Yao et al. (2020). The training datasets are listed in Table 11.

**Evaluation Details.** To demonstrate the generalization capability of each method and assess their practical applicability, we evaluate visual geometry estimation across multiple datasets, including KITTI Geiger et al. (2013), 7-Scenes Shotton et al. (2013), HO3D Hampali et al. (2020), TUM Sturm et al. (2012), ETH3D Schops et al. (2017), HAMMER Jung et al. (2023), and Bonn Palazzolo et al. (2019). For the ablation study, we evaluate on 7-Scenes, Bonn, KITTI, and TUM. Similar to the evaluation of FrozenRecon Xu et al. (2023), each dataset comprises multiple sequences, from which we uniformly sample keyframes for evaluation using a pre-defined stride between consecutive frames, with a maximum keyframe number of 200. We perform evaluations on the NVIDIA H200 GPU. Due to CUDA memory constraints, we limit the number of frames per sequence to a maximum of 200. The evaluation datasets are listed in Table 12. For KITTI, we use the sequences of 2011_09_26_0001, 2011_09_26_0009, 2011_09_26_0091, 2011_09_28_0001, 2011_09_29_0004, and 2011_09_29_0071.

For point cloud estimation, we aggregate the predictions of each sequence in world coordinates by stacking the individual estimations. To assess both per-view accuracy and cross-view consistency, we align the stacked predicted point cloud with the corresponding stacked ground-truth point cloud using a similarity transformation comprising a scale factor, a rotation matrix, and a translation vector.

For camera pose translation vector and video depth estimation, we align the predictions with the ground truth through a scale value for each sequence. For monocular depth estimation, we align a scale value for each image.

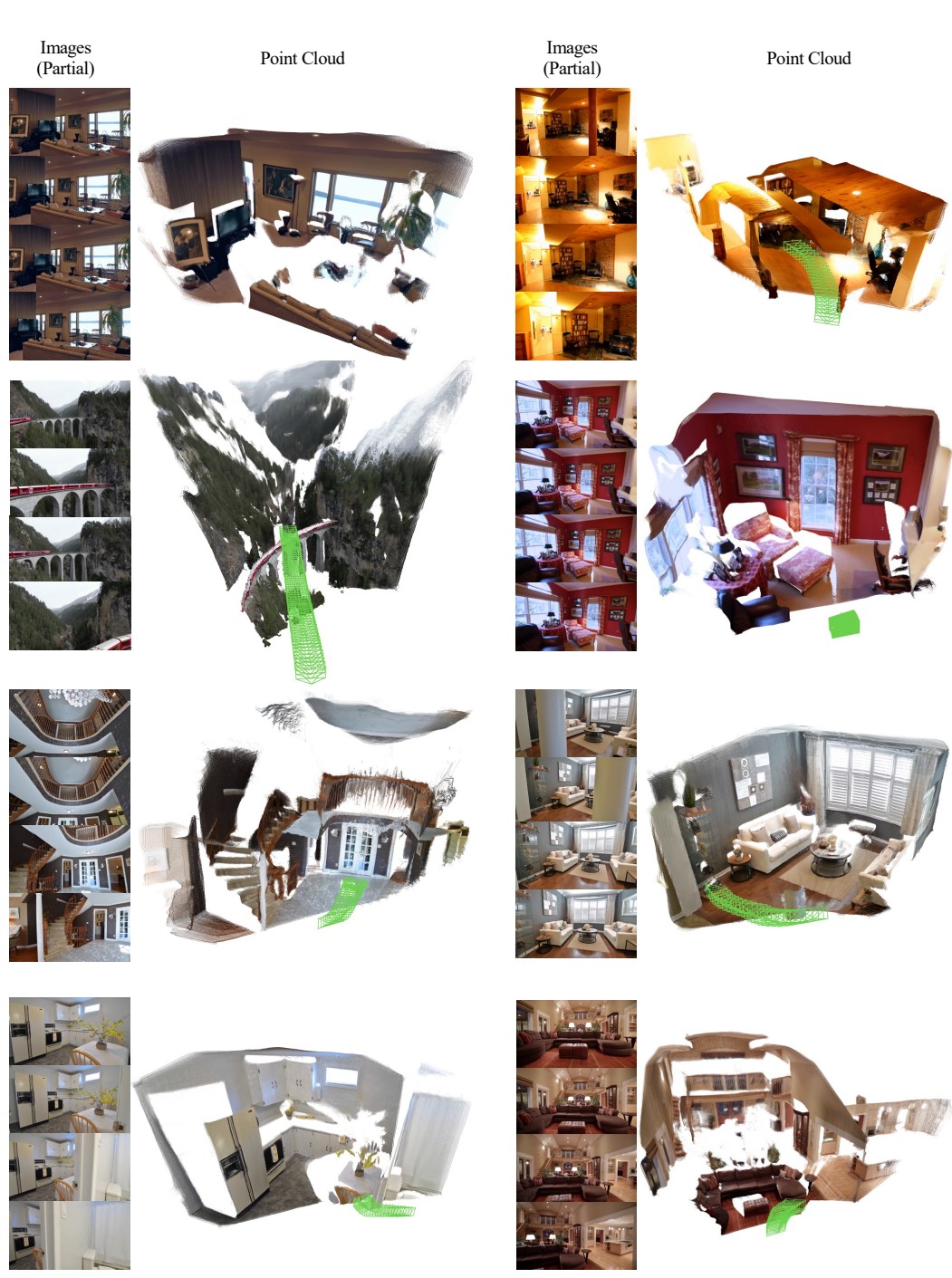

Figure 3: More quantitative results of our CARVE model.

