# OpenReview forum: "CARVE: Dissecting Core Components for Accurate and Resolution-Enhanced Visual Geometry Estimation"
_ICLR.cc/2026/Conference — ICLR 2026 Conference Withdrawn Submission_

### Official Review · Reviewer_UT3W · 2025-10-26

**Soundness:** 3
**Presentation:** 3
**Contribution:** 2
**Rating:** 6
**Confidence:** 3

**Summary:**

This paper systematically dissect the core components to enhance the performance of VGGT. It carefully designs the training procedure, proposes a consistency loss, and, furthermore, integrates high-resolution features into low-resolution features via cross-attention, enabling accurate estimation from high-resolution inputs with less computational burden. Experiments show the proposed CARVE framework achieve SOTA results among multiple tasks.

**Strengths:**

1. This paper improves several core components of VGGT to improve its performance to achieve better performance.
2. It carefully designs the training procedure, proposes a consistency loss, and, furthermore, integrates high-resolution features into low-resolution features via cross-attention, enabling accurate estimation from high-resolution inputs with less computational burden.
3.  Experiments show the proposed CARVE framework achieve SOTA results among multiple tasks.

**Weaknesses:**

1. This paper is a good systematical paper to improve performance of current work, i.e. VGGT, but is with limited technical novelty. Its seems all of the components comes from existing works, and combines them to improve the performance of VGGT.
2. Although this paper claims there are some novel idea, such as consistency loss, but the only evaluation is to apply it to VGGT. It is difficult to know whether the proposed new components are really useful or only can be used in VGGT. It will make the contribution of this paper minor.
3. The "Section 2 Related Work" is too concise and make it not necessary to be a section.

**Questions:**

1. It would be better to figure out what are the new designs to improve the performance? and what are the performance improvement modules from existing works?
2. If there are any new designs proposed by this paper, is any experiment to show its generalization ability?
3. The "Section 2 Related Work" is too concise. If it is because of the page limits, it would be better to be combined with Section Introduction.

---

### Official Review · Reviewer_YV9d · 2025-10-28

**Soundness:** 3
**Presentation:** 3
**Contribution:** 2
**Rating:** 2
**Confidence:** 4

**Summary:**

The paper revisits the VGGT framework and proposes several strategies to enhance its performance. It systematically evaluates different loss formulations and identifies an effective combination that yields the best results. The authors further expand the training data and introduce a strategy for leveraging high-resolution images without significantly increasing memory consumption compared to the original VGGT.

**Strengths:**

- The method leads to improved accuracy compared to VGGT.
- Scales better with high-resolution images than VGGT.

**Weaknesses:**

- The paper primarily presents an extensive ablation study. It appears more as a journal-style extension of VGGT rather than a standalone contribution. While the work is technically sound, it does not introduce substantial new insights. The approach largely revisits existing loss formulations, explores additional datasets, and achieves incremental accuracy improvements over VGGT. The proposed “novel” consistency loss in Eq. 8 is effectively equivalent to the standard transfer error.
- Finding 1 is not particularly insightful, as it merely states that including more diverse scenarios in the training set improves generalization - a well-known observation. It is somewhat odd that such findings are highlighted as the paper’s main conclusions despite their triviality: "diverse data is better than non-diverse data", "high-resolution inputs yield better performance", and similar statements. These read more as straightforward empirical results than conceptual contributions and seem misplaced when introduced so early in the method description.
- MapAnything is missing from the experiments.
- Although the proposed method achieves improvements over VGGT, it cannot be regarded as a genuinely new approach. It mainly revises the combination of losses and training data used in VGGT. The inclusion of both high- and low-resolution images is a reasonable idea, and while it enables processing slightly higher-resolution inputs, the resulting gains reported in Table 1 are modest. This represents a solid incremental improvement rather than being a game-changer.

**Questions:**

I do not really have questions. The paper and its contributions are clear. My main problem is that it is more of a journal-extension of VGGT than a standalone paper.

**Details Of Ethics Concerns:**

No concerns

---

### Official Review · Reviewer_bTqU · 2025-11-01

**Soundness:** 3
**Presentation:** 3
**Contribution:** 3
**Rating:** 6
**Confidence:** 3

**Summary:**

This paper introduces CARVE, a visual geometry estimation framework that systematically analyzes the VGGT architecture and reveals counterintuitive findings: data diversity significantly outperforms data quality in boosting accuracy, while widely used spatial gradient loss and learnable confidence weighting degrade performance, and local region alignment reduces accuracy. Building on these insights, the authors propose two key contributions: a novel consistency loss $L_{\text{consis}}$ that enforces geometric coherence between predicted depth maps, camera parameters, and 3D point clouds via intrinsic

**Strengths:**

S1: The paper uncovers unexpected truths about training components in visual geometry models—showing that data diversity matters more than quality and that common loss designs hurt performance—offering a fresh empirical perspective.

S2: The experiments are exceptionally thorough, spanning seven benchmarks with consistent protocols, careful ablations, and precise evaluation methods that leave little room for doubt.

S3: The explanations are crisp and well-supported: figures and tables clearly convey how the consistency loss and gated feature fusion work, making even technical innovations easy to grasp.

S4: By enabling high-resolution 3D estimation without crippling compute costs, CARVE bridges a key gap between academic models and real-time applications in robotics,自动驾驶, and spatial computing.

**Weaknesses:**

W1: CARVE uses the fixed inverse-depth weighting $W_{\text{inv}}$ without normalization or adaptation to handle starkly different depth distributions across datasets (e.g., long-range KITTI vs. close-range 7-Scenes), yet no analysis is provided on its robustness under such variation.

W2: The consistency loss $L_{\text{consis}}$ relies on predicted intrinsics to compute unprojection, but the paper does not examine how errors in estimated FOV or optical center—measured as FOV Rel in evaluations—propagate into this loss or affect geometric coherence, leaving its sensitivity to calibration noise unaddressed.

W3: While CARVE employs the same training data (Data3) as VGGT, its gains may stem from increased sequence length (up to 50 vs. 24 frames) and batch dynamics rather than the proposed architecture; no ablation isolates the impact of the cross-attention fusion module under identical training conditions, making it unclear how much improvement is architectural versus logistical.

**Questions:**

See weakness.

---

### Official Review · Reviewer_L4hV · 2025-11-01

**Soundness:** 3
**Presentation:** 3
**Contribution:** 1
**Rating:** 2
**Confidence:** 5

**Summary:**

This work presents CARVE, which systematically analyzes key components of the VGGT framework for visual geometry estimation. It explores the effects of data composition, training objectives, and resolution on performance, and proposes two enhancements: a consistency loss to enforce geometric coherence and an efficient high-resolution adaptation module. Extensive experiments are conducted across multiple benchmarks, showing improved accuracy and efficiency over VGGT.

**Strengths:**

* Extensive experiments are conducted to explore the existing components of VGGT.


* Findings are well summarized and may provide insights for enhancing VGGT.


* The paper is well-structured and easy to follow.

**Weaknesses:**

* The observations are tightly coupled to the VGGT architecture. Although CARVE presents several findings, all experiments are conducted solely on VGGT, making it hard to tell whether the observations generalize. If the goal is only to improve VGGT, the contribution is more of an engineering effort than a research advancement.

* The contribution lacks novelty and is not thoroughly justified. The proposed consistency loss is essentially an implementation of an un-projection operation. Besides, enforcing such a loss might contradict the original VGGT motivation of “enhancing overall accuracy despite redundancies”. More in-depth analysis is needed to justify its effectiveness, such as training from scratch, comparing with variants that remove specific prediction heads.

* The improvements appear minor. As shown in Tab. 7 to Tab. 10, CARVE does not consistently outperform VGGT across datasets and metrics, and the gains are small even when improvements occur. Given that CARVE is fine-tuned from VGGT pretrained weights with extra computation and data resources, it is unconvincing that such improvements are significant enough for an ICLR-level contribution.

**Questions:**

In Finding 1, CARVE suggests that “Compared with data quality, data diversity exerts a greater influence”. However, it is unclear how this conclusion is drawn. Data3 includes more data than Data2 (increasing both quality and diversity), and the gains from “Data2 -> Data1” are similar to those from “Data3 -> Data2”, making it difficult to separate the effects of data quality and diversity in these experiments. The authors are suggested to further clarify this finding.

---

### Note · Authors · 2025-11-14

I have read and agree with the venue's withdrawal policy on behalf of myself and my co-authors.